# Size not Language: Effects of External Evidence in Multilingual Medical Question Answering

## Abstract

This paper investigates, for the first time, Multilingual Medical Question Answering across high-resource (English, Spanish, French, Italian) and low-resource (Basque, Kazakh) languages. We evaluate three types of external evidence, such as local repositories, dynamically web-retrieved content, and LLM-generated explanations with models of varying size. Our results show that larger models consistently perform the task better in English for both the baseline evaluations and when adding external knowledge. Interestingly, retrieving the evidence in English often surpasses language-specific retrieval, even for non-English queries. These findings challenge the assumption that language-related external knowledge uniformly improves performance and reveal that effective strategies depend on both the source of language resources and on model scale. Furthermore, specialized static repositories such as PubMed are limited: while they provide authoritative expert knowledge, they lack adequate multilingual coverage and do not fully address the reasoning demands of the task. Code and resources publicly available: `https://anonymous.4open.science/r/multilingual-medical-qa-C635/`

## 1 Introduction

The rapid advancements in Large Language Models (LLM) research have yielded impressive results across various domains, including healthcare (Brown et al., 2020; Achiam et al., 2023; Liévin et al., 2024). LLMs demonstrate strong capabilities in clinical reasoning and decision-making across tasks of varying complexity, opening the door to potential applications in real-world medical contexts.

Nevertheless, despite the continuously increasing functionalities of the LLMs, they still struggle with hallucinations, limited context length, and ungrounded generation, which undermine their consistency and factual accuracy, potentially causing serious harm in critical domains such as medicine (Ahmad et al., 2023; Yu et al., 2023; Kim et al., 2025; Artsi et al., 2025; Roustan et al., 2025).

To mitigate these challenges, several strategies have been explored, including training specialized medical models (Sellergren et al., 2025; Zhang et al., 2023; Chen et al., 2024), curating more diverse datasets (Jin et al., 2021; 2019; Pal et al., 2022), and grounding outputs in established knowledge sources (Xiong et al., 2024a; Alonso et al., 2024; Biesheuvel et al., 2025). Among these, relying on other sources through methods such as Retrieval-Augmented Generation (RAG) has become especially prominent, as it helps compensate for the gaps in LLMs' internal knowledge. Yet, most of this work is concentrated on English, making it difficult to generalize findings across other languages. For instance, Alonso et al. (2024) highlights a stark performance drop for French, Italian, and Spanish compared to English, even when using RAG, indicating that non-English medical Question Answering (QA) remains significantly under-researched. Additionally, almost all the databases with medical expert knowledge are in English (Xiong et al., 2024a; Amugongo et al., 2025), which makes it unclear how to deal with medical exams in settings in which the target language is a language other than English.

At the same time, scaling up LLMs has shown that they can encode a surprising amount of specialized domain knowledge within their parameters (Brown et al., 2020; Ren et al., 2023). This highlights the need to better understand how to combine intrinsic domain knowledge with external resources to

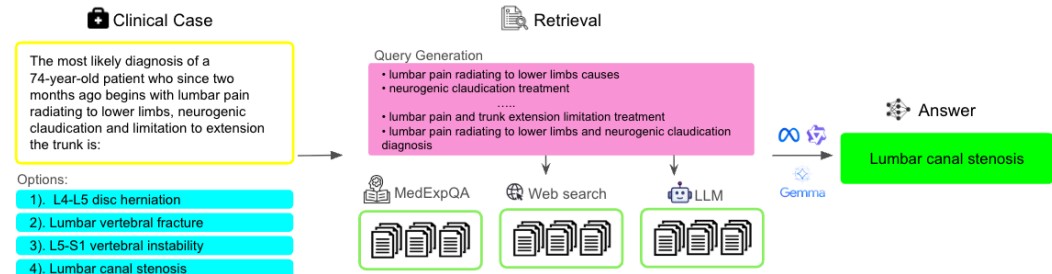

Figure 1: Description of the Multilingual Medical Question Answering pipeline. First, given a question related to a clinical case and the options to answer it, we generate 10 queries taking as a basis the clinical case. Next, using these queries, we obtain dynamically external knowledge from: i) the web, and ii) generating the answers using LLMs. Pre-retrieved passages from the MedExpQA dataset are also used to obtain static information related to the clinical case. Finally, we provide these documents to some state-of-the-art LLMs and generate the answer.

achieve optimal performance. Taking this into consideration, we formulate the following research questions:

- Is there a single method that consistently performs best across all languages?

- How do the retrieval quality and generation accuracy with LLMs compare when using English as a source versus those that incorporate these techniques across other target languages?

- Do LLMs encode enough external knowledge for optimal performance in Multilingual Medical Question Answering?

Our findings show several important insights. Firstly, using LLMs with web search in English is the best strategy for Multilingual Medical Question Answering in any language. Second, larger models consistently demonstrate superior performance compared to smaller models across all evaluated settings. Third, and most importantly, the performance disparity between high-resource and low-resource languages is primarily influenced by two factors: model scale and the language of the external information sources. Notably, methods using evidence retrieved in English frequently yield better results than retrieval conducted in the target language, even for non-English queries. This may explain the fact that web search in English is also beneficial in other target languages. Lastly, locally used and traditionally reliable repositories of medical information, such as PubMed, exhibit incomplete coverage of the medical domain, both in general and across different languages.

Our results challenge a common assumption in knowledge-augmented medical Question Answering: that adding external evidence uniformly improves performance. Instead, our multilingual experiments highlight complex dynamics that vary across languages and model sizes, showing that retrieval strategies cannot be one-size-fits-all. To investigate these dynamics, we examine three types of external evidence: (1) medical local knowledge repositories such as PubMed or Wikipedia, (2) web-retrieved documents, and (3) LLM-generated knowledge. Our approach is illustrated in Figure 1. We evaluate how each type of evidence interacts with models of different sizes in multilingual medical QA tasks, and we report detailed findings on their relative effectiveness. In this study, we contribute the following:

- A systematic analysis of external knowledge integration strategies for medical QA across high-resource languages (English, Spanish, French, Italian) and low-resource languages (Basque, Kazakh).

- Empirical evidence showing how model size influences the effectiveness of knowledge augmentation in different linguistic contexts.

- Identification of the most effective retrieval and integration strategies across languages and model sizes.

## 2 RELATED WORK

The application of LLMs in the medical domain is one of the crucial directions in the development of AI. The growth of the model sizes along with their parameter counts suggests that current language models encode a significant amount of specialized knowledge (Wei et al., 2022; Singhal et al., 2023). This capability has sparked extensive research into evaluating how well state-of-the-art LLMs perform on specialized clinical tasks and medical knowledge domains.

Recent research has systematically assessed the medical competency of prominent foundation models, including Llama (Touvron et al., 2023; Grattafiori et al., 2024), Mistral (Jiang et al., 2024), Gemma (Team et al., 2025), and Qwen (Bai et al., 2023), among others. While large language models (LLMs) exhibit strong performance in medical knowledge recall and clinical reasoning, significant challenges persist in ensuring reliability, detecting hallucinations, managing rare conditions, and maintaining consistency with established clinical guidelines.

**Medical QA**. Several datasets have been constructed with the explicit aim of evaluating these limitations. Jin et al. (2019) introduced PubMedQA, a biomedical question-answering dataset that includes a question extracted from the title, a context that summarizes the abstract, a long answer or conclusion of the abstract, and a final decision classified as *Yes*, *No* or *Maybe* based on the abstract's conclusion. Jin et al. (2021) introduced MedQA a multiple-choice question benchmark derived from real medical licensing exams, such as the USMLE and equivalent exams from other countries (e.g., China or India in extended versions), designed to assess whether a model can answer complex clinical questions that require not only factual knowledge but also clinical reasoning. Pal et al. (2022) designed MedMCQA a medical multiple-choice question answering dataset written in English and specifically focused on real medical entrance exam questions in India, containing questions spanning 2,400+ healthcare topics across 21 medical subjects, including anatomy, pharmacology, pathology, and clinical disciplines. Originally in Spanish, Agerri et al. (2023) created a parallel multilingual dataset from Spanish Resident Medical Intern exams that later, and with the help of the *CasiMedicos* volunteer doctor community, was enriched with explanations to the correct and incorrect answers.

**RAG in the medical domain.** To address the issues of knowledge scarcity in some medical areas and questions, methods such as Retrieval Augmented Generation (RAG) have been explored (Lewis et al., 2020; Gargari & Habibi, 2025). For instance, the model architecture MedRAG (Xiong et al., 2024a) comes as a part of a framework for medical RAG. MKRAG (Shi et al., 2025) incorporates fact-based retrieval from external medical knowledge bases and demonstrates a 4% improvement in accuracy on the MedQA benchmark. MedExpQA (Alonso et al., 2024) is a benchmark designed to evaluate medical question-answering systems that leverage external knowledge sources, with a focus on multilingual capabilities to assess performance across different languages.Xiong et al. (2024b) proposed a retrieval system that iteratively improves search results by incorporating continuous follow-up questions, enabling the system to refine its understanding and provide more accurate and context-aware information retrieval.

## 3 KNOWLEDGE SOURCES

This section details the methodology adopted to answer our research questions. We intend to encounter an optimal strategy and analyze their influence to the final decision between different sources to obtain additional information for the medical domain and define the most effective ones: well-established medical knowledge repositories (Section 3.1), information from the web (Section 3.2), or the parametric knowledge encoded in LLMs (Section 3.3).

### 3.1 RETRIEVING LOCAL KNOWLEDGE

**MedExpQA** (Alonso et al., 2024). In the MedExpQA benchmark for each question a collection of 32 related documents was retrieved from multiple knowledge sources, including PubMed, Wikipedia, StatPearls, and medical textbooks. As the knowledge storage is predominantly in English and the most relevant and informative documents are in this language, in our experiments, we used the English subset of the retrieved documents. Retrieved data are ranked by relevance, where the highest-ranked document corresponds to the highest similarity score with respect to the clinical case question.

## 3.2 Web Search

**Search Query Generation**. To obtain relevant evidence via web search and LLM, we first generate 10 queries per question describing the clinical case using *Llama-3.3-70B-Instruct*, ensuring no overlap with the models used in the main evaluation (Panickssery et al., 2024). These queries describe the clinical case in a way that helps locate useful information for answering the question, without directly pointing to the correct answer. Consequently, the generated queries are used for the web search engine to retrieve candidate evidence passages. For cross-lingual comparability, we first generate the queries in English and then translate them, using the same model, to the rest of the languages of the dataset.

**Search APIs**. As an additional external source, we retrieve documents from the web using two APIs: the web search tool by Cohere API [1] and Google Search using Serper API [2]. The former provides both retrieved documents and a summary generated by its underlying language model in response to the query, resulting in a more comprehensive answer. However, the retrieved content and generated outputs may be subject to limitations imposed by the model itself. To mitigate such constraints and enable broader access to web content, we also employ the Serper API, which offers direct access to Google Search results without interference, filtering or any summarization by any LLM. We refer to the search results from one query as 1 document, resulting in 10 documents to mean results obtained from 10 queries.

We use the generated search queries as input for the APIs and retrieve the information using the queries independently for each language. Nevertheless, the imbalance of the information available on the web for less-resourced languages is evident, and we report the ratio of the retrieved data per language in Table 1.

Table 1: Comparison of retrieved document availability between web-search using Google Search and Cohere across languages.

| Metric | English | Spanish | Italian | French | Basque | Kazakh |
|---|---|---|---|---|---|---|
| *Average documents retrieved per query* | | | | | | |
| Google Search | 9.24 | 9.29 | 9.32 | 9.32 | 2.54 | 4.17 |
| Cohere | 4.12 | 3.96 | 3.90 | 3.88 | 2.43 | 2.61 |
| *Percentage of empty sources (no results found)* | | | | | | |
| Google Search | 0% | 0% | 0.08% | 0% | 54.8% | 31.36% |
| Cohere | 1.36% | 1.2% | 2.32% | 1.92% | 17.68% | 22.08% |

## 3.3 Parametric knowledge of LLMs

Recent models have demonstrated that current language models could yield impressive performance on many tasks, including medical exams. Our initial results show that the performance of the LLMs excels in medical QA in English without any additional information, which leads us to hypothesize that these LLMs pre-trained on an immense amount of world knowledge encode sufficient information in the medical domain within their parameters. Hence, in order to answer this question, we ask LLMs to generate answers to the queries used for web search and generate an answer directly from the clinical case question. This step is achieved in two ways: (1) we provide the input question describing a clinical case, and prompt a model to generate an explanation of the correct answer for the question directly, and (2) for each of the search queries generated for the web search, we prompt models to generate their answers to these queries. We generate the explanations both in English and in the target language.

---

[1] https://cohere.com/
[2] https://serper.dev/

## 4 EXPERIMENTAL SETUP

### 4.1 DATA

In our experiments, we use the CasiMedicos dataset Agerri et al. (2023), a multilingual parallel medical dataset of commented medical exams. This dataset comprises comprehensive clinical case scenarios presented as multiple-choice questions, where each question includes a detailed clinical case description, a set of potential diagnostic or treatment options, and the corresponding correct answer. The original CasiMedicos dataset provides parallel translations across four languages: English, Spanish, Italian, and French. The data consists of the 125 questions in the test set that we use in our experiments.

Recognizing that the four languages included in the original CasiMedicos corpus represent relatively high-resource languages from the same linguistic family with extensive digital presence and computational support, we sought to extend our analysis to encompass lower-resource linguistic contexts. We additionally translate the dataset into two lower-resource languages, Basque and Kazakh, using Claude-4-Sonnet[3]. Both languages represent different linguistic families, with the latter additionally representing a different writing script (Cyrillic).

Such a multilingual coverage allows for a comprehensive evaluation of external knowledge integration strategies across a spectrum of language-specific resource availability, from well-supported higher-resource languages to computationally underrepresented language families.

### 4.2 MODELS

To systematically evaluate the influence of different types and levels of external knowledge sources described in the preceding sections, we conducted comprehensive experiments using a diverse set of state-of-the-art language models. Our experimental setup included the following models:

- **Qwen** (Bai et al., 2023): the 8B, 14B, and 32B parameter versions from Qwen 3, and the 72B parameter instruction-tuned model from Qwen 2.5.
- **Llama** (Touvron et al., 2023): Llama3.1-Instruct models with 8B and 70B parameters.
- **Gemma** (Team et al., 2025): Instruction-tuned version of Gemma models with 12B and 27B parameters, and *MedGemma*(Sellergren et al., 2025) with 27B parameters.

Although the language coverage between Qwen-2.5 (29 languages) (Yang et al., 2024; Team, 2024) and Qwen-3 (100+ languages) (Team, 2025) is substantial, we use only the 72B model of Qwen-2.5 to compare it with 70B version of the Llama model in high-resource languages (English, Spanish, Italian and French).

The goal of the model was to analyze the question, provided documents that help to find the correct answer and choose the correct option. This experimental design enables us to investigate the trade-off between information quality and quantity in retrieval-augmented generation scenarios in the medical domain. Specifically, we aim to determine whether providing more contextual documents consistently improves performance or whether there exists an optimal balance point where additional information begins to introduce noise that degrades model accuracy.

The documents from every source differ in their length and content. In case of MedExpQA, the documents are the files with reports, analysis and general definitions. In LLM-generated and web-search, we get the precise answer directly addressing the search query, which is short and concise in its nature.

## 5 RESULTS AND ANALYSIS

In this section, we describe the results after conducting the experiments described in the previous section. We report the accuracy of the predictions from different models. We explain the model performance analysis, describe observed performance trends and define the best performing method based on the results.

---

[3]https://www.anthropic.com/claude/sonnet

Table 2: Performance comparison across different retrieval methods, model sizes and languages. The models are grouped by the parameter size. The results under the orange row are the results from retrieval in English and the blue row is the results from retrieval in the language of the question. The **underlined and bold results** shows the best results per language (row), and **bold text** means the best performing model per parameter count (column). The ↓ indicates the drop in performance and = indicates no improvement compared to the baseline. Underlined results correspond to the best results for each language in a given retrieval setting.

| Params | 8B | | 12B-14B | | 27-32B | | | >=70B | | Avg (Std.) |
|---|---|---|---|---|---|---|---|---|---|---|
| Models | LLaMA 8B | Qwen 8B | Gemma 12B | Qwen 14B | Gemma 27B | MedGemma 27B | Qwen 32B | LlaMA 70B | Qwen 72B | |
| **Baseline: no-retrieval** | | | | | | | | | | |
| EN | 61.6 | 69.6 | 63.2 | 72.0 | 70.4 | 76.8 | 80.8 | 76.0 | **84.0** | 72.7 (7.07) |
| ES | 50.4 | 65.6 | 60.8 | 64.8 | 72.8 | 75.2 | 76.0 | 80.0 | 79.2 | 69.4 (9.24) |
| FR | 53.6 | 58.4 | 62.4 | 67.2 | 73.6 | 75.2 | 74.4 | **83.2** | 81.6 | 69.9 (9.65) |
| IT | 48.8 | 62.4 | 68.8 | 67.2 | 68.0 | 72.0 | 75.2 | 79.2 | 81.6 | 69.2 (9.22) |
| EU | 36.8 | 45.6 | 48.0 | 57.6 | 62.4 | 67.2 | 57.6 | 71.2 | 53.6 | 55.6 (10.26) |
| KK | 36.8 | 44.0 | 54.4 | 42.4 | 53.6 | 63.2 | 64.8 | 72.8 | 57.6 | 54.4 (11.04) |
| Avg. (Std.) | 48.0 (8.88) | 57.6 (9.67) | 59.6 (6.69) | 61.87 (9.71) | 66.8 (6.96) | 71.6 (4.88) | 71.47 (7.82) | 77.07 (4.17) | 72.93 (12.39) | |
| **MedExpQA** | | | | | | | | | | |
| EN | 72.8 | 77.6 | 67.2 | 76.8 | 74.4 | 70.4↓ | 80.0↓ | 79.2 | 81.6↓ | 75.56 (4.48) |
| ES | 62.6 | 68.8 | 70.4 | 72.0 | 79.2 | 76.8 | 76.8 | 80.0= | 80.8 | 75.16 (5.76) |
| FR | 67.2 | 68.8 | 68.0 | 75.2 | 78.4 | 76.8 | 78.4 | 79.2↓ | 80.0↓ | 74.58 (4.87) |
| IT | 64.0 | 70.4 | 63.2↓ | 73.6 | 75.2 | 77.6 | 79.2 | 81.6 | 80.8↓ | 73.96 (6.46) |
| EU | 44.8 | 59.2 | 63.2 | 58.4 | 67.2 | 73.6 | 67.2 | 73.6 | 67.2 | 63.82 (8.42) |
| KK | 45.6 | 56.0 | 56.0 | 61.6 | 70.4 | 70.4 | 69.6 | 71.2↓ | 61.6 | 62.49 (8.32) |
| Avg. | 59.5 (10.61) | 66.8 (7.21) | 64.67 (4.65) | 69.6 (7.01) | 74.13 (4.22) | 74.13 (2.91) | 75.2 (4.95) | 77.47 (3.74) | 75.33 (7.91) | |
| **Generated explanations** | | | | | | | | | | |
| EN | 67.2 | 73.6 | 69.6 | 75.2 | 67.2↓ | 68.8↓ | 75.2↓ | 76.0= | 80.0↓ | 72.53 (4.25) |
| ES | 63.2 | 70.4 | 66.4 | 74.4 | 65.6↓ | 74.4↓ | 74.4↓ | 79.2↓ | 78.4↓ | 71.82 (5.39) |
| FR | 68.0 | 70.4 | 60.0↓ | 73.6 | 64.8↓ | 72.8↓ | 75.2 | 76.8↓ | 80.0↓ | 71.29 (5.87) |
| IT | 68.0 | 71.2 | 66.4↓ | 78.4 | 64.8↓ | 74.4 | 79.2 | 80.0 | 80.8↓ | 73.69 (5.91) |
| EU | 53.6 | 60.8 | 65.6 | 62.4 | 65.6 | 71.2 | 71.2 | 76.0 | 69.6 | 66.22 (6.33) |
| KK | 61.6 | 58.4 | 58.4 | 59.2 | 66.4 | 66.4 | 72.0 | 72.0↓ | 63.2 | 64.18 (5.07) |
| Avg. (Std.) | 63.6 (5.09) | 67.47 (5.71) | 64.4 (3.91) | 70.53 (7.10) | 65.73 (0.85) | 71.33 (2.94) | 74.53 (2.59) | 76.67 (2.59) | 75.33 (6.62) | |
| *websearch* | | | | | | | | | | |
| EN | 72.0 | 80.0 | 74.4 | 81.6 | 80.0 | 73.6↓ | 78.4↓ | 80.0 | 83.2↓ | 78.13 (3.66) |
| ES | 69.6 | 75.2 | 78.4 | 80.0 | **84.8** | 81.6 | 80.0 | 81.6 | 80.0 | 79.02 (4.13) |
| FR | 71.2 | 75.2 | 76.0 | 77.6 | 80.0 | 78.4 | 79.2 | 81.6↓ | 82.4 | 77.96 (3.27) |
| IT | 72.0 | 78.4 | 76.0 | 81.6 | 79.2 | 80.0 | 77.6 | 81.6 | 82.4 | 78.76 (3.08) |
| EU | 60.8 | 66.4 | 72.0 | 67.2 | 73.6 | **77.6** | 68.8 | 74.4 | 64.8 | 69.51 (5.01) |
| KK | 60.0 | 62.4 | 70.4 | 68.8 | 72.0 | **77.6** | 70.4 | 73.6 | 68.0 | 69.24 (5.09) |
| Avg. (Std.) | 67.6 (5.16) | **72.93 (6.37)** | 74.53 (2.67) | **76.13 (5.92)** | **78.27 (4.29)** | **78.13 (2.47)** | **75.73 (4.42)** | **78.8 (3.45)** | 76.8 (7.48) | |
| **Generated explanations** | | | | | | | | | | |
| ES | 58.4 | 70.4 | 71.2 | 72.8 | 71.2 | 78.4 | 75.2↓ | 73.6 | 76.0 | 71.91 (5.37) |
| FR | 62.4 | 69.6 | 65.6 | 71.2 | 71.2↓ | 75.2= | 75.2 | 79.2↓ | 78.4 | 72.0 (5.31) |
| IT | 64.8 | 72.8 | 68.8 | 75.2 | 73.6 | 78.4 | 77.6 | 78.4↓ | 80.8 | 74.49 (4.84) |
| EU | 52.0 | 60.8 | 63.2 | 61.6 | 71.2 | 74.4 | 65.6 | 72.8 | 59.2 | 64.53 (6.84) |
| KK | 46.4 | 53.6 | 60.0 | 57.6 | 64.0 | 67.2 | 64.0↓ | 68.8↓ | 65.6 | 60.8 (6.82) |
| Avg. (Std.) | 56.8 (6.77) | 65.44 (7.18) | 65.76 (3.96) | 67.68 (6.84) | 70.24 (3.26) | 74.72 (4.09) | 71.52 (5.58) | 72.0 (8.24) | 74.56 (3.83) | |
| *websearch* | | | | | | | | | | |
| ES | 64.0 | 74.4 | 72.8 | 75.2 | 78.4 | 79.2 | 79.2 | 80.0= | 77.6↓ | 75.64 (4.73) |
| FR | 64.8 | 73.6 | 72.0 | 72.0 | 73.6= | 78.4 | 78.4 | 79.2↓ | 75.2↓ | 74.13 (4.22) |
| IT | 66.4 | 72.8 | 71.2 | 76.8 | 78.4 | 75.2 | 79.2 | 80.0 | 78.4↓ | 75.38 (4.23) |
| EU | 47.2 | 52.0 | 61.6 | 56.8↓ | 63.2 | 69.6 | 62.4 | 68.0↓ | 59.2 | 60.0 (6.78) |
| KK | 48.0 | 32.0↓ | 50.4 | 35.2↓ | 59.2 | 61.6↓ | 35.2↓ | N/A | 41.6↓ | 40.36 (17.38) |
| Avg. (Std.) | 58.08 (8.59) | 60.96 (16.73) | 65.6 (8.62) | 63.2 (15.69) | 70.56 (7.94) | 72.8 (6.54) | 66.88 (17.09) | 61.44 (31.06) | 66.4 (14.24) | |

## 5.1 Overall results

The experimental results in Table 2 show significant performance variations across different retrieval methods, model sizes, and languages. Analyzing the comprehensive evaluation across six languages (English (EN), Spanish (ES), French (FR), Italian (IT), Basque (EU), and Kazakh (KK)), several key patterns emerge regarding optimal configurations for multilingual information retrieval.

**Model Performance.** Intuitively, the main observation from the results is that model size significantly impacts performance, with a general trend of increasing accuracy as the number of parameters grows. The average performance across all retrieval methods and benchmarks shows a clear scaling effect. The largest models mainly exhibit the best performance across all the settings. Qwen-2.5-72B consistently outperforms the 70B version of Llama in the higher-resource languages, and

underperforms in lower-resource languages, which is reflected in the high standard deviation. It happens due to poor multilingual representation in the 2.5 version of Qwen compared to other models.

Additionally, the decrease in accuracy for Kazakh and Basque can be explained by the influence of the tokenization algorithms optimized for Latin scripts, which demonstrate significantly reduced efficiency when applied to Kazakh and Basque text, as the Cyrillic-based writing system and agglutinative morphology result in extensive fragmentation into smaller subtokens. This increased token density impacts memory consumption, with Kazakh and Basque text requiring substantially more tokens per semantic unit compared to English, thereby increasing both embedding storage requirements and computational overhead in transformer architectures. Therefore, leads to an out-of-memory error for Kazakh with Llama 70B when extra information is added to the model input.

Regarding mid-size models, the MedGemma-27B model shows remarkable performance, often matching or exceeding larger models, especially in lower-resource languages. Among all the models, MedGemma shows the most robust performance across all the languages with the least variation in its results.

Qwen 32B demonstrates consistent, strong performance across all evaluation conditions, often equaling the performance of the bigger models. Nevertheless, it often falls short in two lower-resource scenarios. Finally, In the group of the smaller models, Qwen 8B consistently outperforms all other models of a similar size.

**Performance Trends.** Despite achieving strong baseline performance, the incorporation of additional external data frequently leads to performance degradation rather than improvement in high-resource languages compared to the baseline results. This trend is particularly pronounced in larger models, suggesting a negative correlation between model size and the beneficial utilization of external data sources. Nevertheless, the opposite trend is observable with the two less-resourced languages.

On the other hand, when comparing with the results per source of external data, we still witness the same trend outlined previously: the larger the model, the better the performance. Therefore, we may suggest that LLMs with a bigger parameter count encode sufficient information and additionally, retrieved information introduces noise

Nevertheless, the hypothesis that larger models encode more domain-specific knowledge in their parameters is rejected by the performance of the LLM-generated evidence. Although this can be an effective strategy, it is not as powerful as external retrieval. Hence, smaller models still benefit from this method, especially in lower-resourced languages.

**Method Performance.** Among all external knowledge integration strategies evaluated, optimal performance was consistently achieved when clinical case questions were augmented with retrieved information from web-based sources in English, with a minor gain in results obtained with the Cohere API, which is illustrated in Table 2. The Cohere API provides LLM-generated summaries that synthesize information from multiple retrieved sources alongside direct web-extracted content.

Such performance can be attributed to the query-specific nature of web-based retrieval systems, which excel at identifying information that directly addresses the posed search query. In contrast to static document collections, web-search engines dynamically identify and prioritize content with high semantic relevance to the immediate query context. This distinction becomes particularly significant when compared to pre-defined knowledge bases such as MedExpQA, where document retrieval relies on similarity metrics that may identify topically related but not always directly applicable content.

## 5.2 SEARCH LANGUAGE

Even when information is scarce in certain target languages, we still conduct retrieval operations using web search in those specific languages rather than defaulting to English. The results from this approach show considerable variation in performance quality across different languages. However, the key finding that emerges from this analysis is that system performance consistently falls short of optimal levels when the retrieved contextual information is presented in other languages rather than English.

Moreover, the gap in the retrieval content is more evident in Google Search. In contrast, when Cohere API fails to locate relevant information in the originally requested language, its underlying LLM automatically implements a hierarchical language fallback approach to maximize retrieval success.

For instance, when processing a query in Basque, the system first attempts retrieval in the original Basque query. If this initial search yields insufficient or no relevant results, the system automatically translates the query into Spanish, leveraging the linguistic and cultural proximity between these languages as well as Spanish's significantly larger digital content repository. Should the Spanish-language search also fail to produce satisfactory results, the system escalates to English as the final fallback language.

## 5.3 MULTILINGUAL SEARCH RESULTS

We experiment with the monolingual setting when the context documents are in the language of the question. The results are shown in Table 2 under the rows in blue. Here, the observations made from the English search results are more evident. No matter the parameter size of the models, we conclude that there are no major improvements in the performance. Larger models generally perform better than smaller ones. However, with in-language documents, the gains from increasing model size are less dramatic than when the retrieved documents are in English. Moreover, the gap between higher and lower resourced languages is more evident, which could be explained primarily by the lack of sufficient information in the language in the pre-training data and the external knowledge bases.

## 6 DISCUSSION

### 6.1 ANALYSIS OF THE SOURCE

Based on the results shown in Section 5, we can conclude that smaller <30B parameter models benefit from the data provided from outside sources. Nevertheless, the obvious gain is obtained for English and less frequently for French, Spanish and Italian. For models with more than 12B parameters, results obtained through web searches in English begin to approach the best results over the baseline for Basque and Kazakh.

Despite these gains, overall, we see how the accuracy of the correct answer drops more frequently in the large models, indicating that either no external knowledge is beneficial or the larger the models, the more confident they become in their internal parametric knowledge.

Several factors, such as length of the context, number of documents and the retrieval algorithm, the quality of the retrieved data can influence such behaviour. In the case of a difference between web retrieval, the amount of information extracted from each retrieval approach varies in its underlying search algorithm. For instance, Cohere API tries to extract the 5 most relevant sources for each query. Serper API provides the 10 best-matching sources. LLM-generated responses return 1 paragraph-level answer per query. MedExpQA has a set of 32 relevant documents of varying size for each question.

When it comes to answering the question of whether the information in the trusted expert data repositories, such as PubMed and Wikipedia, the documents provided in MedExpQA, are enough, we additionally look into the retrieved sources from the web search. In our analysis, we can see that web-search sources cover the data stores from the sources of MedExpQA. On average, 11.2% of the information was retrieved from Wikipedia and 18.8% of the information was retrieved from PubMed, when retrieved in English.

### 6.2 COMPARISON WITH OTHER MEDICAL BENCHMARKS

To strengthen the conclusion made from our experiments, we additionally performed the same set of experiments across three other prominent medical benchmark English datasets: MedQA, PubMedQA, and MedMCQA, using web search (WS) as the external knowledge source based on the performance gain it provides in our previous experiments.

The results in Table 3 demonstrate a consistent pattern of improvement across all benchmarks and model sizes. Across the 8-14B parameter models, we observe substantial gains: Llama3.1-8B shows improvements of 8.17, 9.54, and 4.2 points on MedQA, MedMCQA, and PubMedQA, respectively, while Qwen3-14B demonstrates gains of 6.58, 7.90, and 5.6 points. The 27-32B models exhibit similar trends, with Qwen3-32B achieving a notable 9.5-point improvement on MedQA, and even the medical-specific MedGemma-27B showing consistent gains despite its domain fine-tuning. The

Table 3: Performance comparison of language models with and without retrieved external data (WS) across English Medical QA benchmarks

| | MedQA | | MedMCQA | | PubMedQA | |
|---|---|---|---|---|---|---|
| Model | Base | +WS | Base | +WS | Base | +WS |
| *8-14B* | | | | | | |
| Llama3.1-8B | 56.17 | 64.34 | 55.10 | 64.64 | 70.4 | 74.6 |
| Qwen3-8B | 58.44 | 67.71 | 56.42 | 65.50 | 72.4 | 71.0 |
| Gemma3-12B | 56.01 | 61.04 | 51.23 | 54.75 | 66.0 | 76.4 |
| Qwen3-14B | 63.01 | 69.59 | 56.89 | 64.79 | 72.0 | 77.6 |
| *27-32B* | | | | | | |
| Gemma3-27B | 61.74 | 65.75 | 59.72 | 63.35 | 70.6 | 68.0 |
| MedGemma-27B | 67.95 | 70.46 | 61.85 | 66.91 | 70.0 | 73.8 |
| Qwen3-32B | 63.79 | 73.29 | 64.79 | 56.99 | 74.4 | 76.8 |
| *70-72B* | | | | | | |
| Llama3.1-70B | 72.82 | 76.12 | 69.73 | 74.09 | 77.6 | 79.6 |
| Qwen2.5-72B | 70.93 | 74.16 | 66.82 | 72.84 | 76.7 | 78.4 |

large 70-72B models, while showing smaller absolute improvements due to their higher baseline performance, still demonstrate consistent gains across all benchmarks.

# 7 CONCLUSION

This work systematically examined the role of external knowledge integration in Multilingual Medical Question Answering, analyzing a range of retrieval strategies across languages of differing resource availability and models of varying scale. Although the optimal strategy for Multilingual Medical QA would be to use LLMs with web search,, the results demonstrate that the interaction between model size, retrieval source, and language resource status is neither uniform nor straightforward. While smaller and mid-sized models generally benefit from the inclusion of additional evidence, especially when retrieved dynamically from the web, larger models often exhibit diminished or even negative returns. This suggests that beyond a certain scale, external evidence may conflict with or disrupt the internalized domain knowledge encoded during pretraining.

A further central observation is that the choice of retrieval language strongly conditions performance. English-based retrieval, even when applied to non-English queries, consistently outperformed retrieval in the target language, reflecting both the breadth of English medical resources and the scarcity of structured and web-based material in lower-resourced languages. These disparities underscore the persistent limitations faced by multilingual medical QA systems and suggest that language resource availability remains a dominant factor shaping outcomes.

Comparisons across retrieval sources reveal additional nuances. Static collections such as MedExpQA, while reliable and domain-specific, provide incomplete coverage and exhibit diminishing returns when applied across multiple languages. In contrast, web-retrieved evidence offered broader coverage and higher relevance to query-specific contexts, but with greater variability and sensitivity to noise. LLM-generated explanations occupy a middle ground, demonstrating utility for smaller models and lower-resource languages but providing less consistent advantages as model size increases.

Collectively, these results challenge the prevailing assumption that external knowledge augmentation invariably improves LLM performance. Rather, they demonstrate that the efficacy of external evidence depends on the complex interactions between model architecture, retrieval methodology, and linguistic context. This research underscores that multilingual medical question-answering systems demand tailored approaches that account for both linguistic disparities and resource availability constraints, while considering the scaling characteristics of the underlying models.

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
