# OpenReview forum: "Consider Size not Language: Effects of External Evidence in Multilingual Medical Question Answering"
_ICLR.cc/2026/Conference — ICLR 2026 Conference Withdrawn Submission_

### Official Review · Reviewer_tpyq · 2025-10-27

**Soundness:** 2
**Presentation:** 4
**Contribution:** 2
**Rating:** 6
**Confidence:** 4

**Summary:**

This paper presents the first study on MMQA across both several well-studied languages (English, Spanish, French, Italian) and low-resource ones (Basque, Kazakh). The authors evaluate models of different sizes using three types of external evidence—local repositories, dynamically retrieved web content, and LLM-generated explanations. Authors show, across results, that larger models perform best in English, both with and without added external knowledge. Notably, retrieving evidence in English often outperforms language-specific retrieval, even for non-English queries. These results challenge the idea that external knowledge in the query’s language always improves performance. The study also finds that while expert repositories like PubMed provide reliable knowledge, they lack multilingual depth and do not fully support the complex reasoning required by MMQA tasks.

**Strengths:**

1) Code and resources are publicly available.

2) The proposed study allows for the study of different levels of language coverage, and also offers an original view comparing the use of resources (web vs. local), compared to models without external resources. Local use is the most interesting from my point of view, with in particular the idea of ​​using private data (e.g. patient records) which must remain local.

3) This systematic comparison allows the choice of models and strategy to be adapted according to material and resource constraints.

**Weaknesses:**

1) Part of the study is quite classic, especially everything concerning the study of the size of the models.

2) The benchmark ultimately appears to be a translation of existing data, which is a priori unverified: translation errors can be numerous, particularly for languages ​​with few resources, which adds a significant bias to the results.

3) Few models have ultimately been studied, in particular models adapted to the medical field (eg MedGemma), or even open-source models (eg Apollo or OLMo). Similarly, no reasoning model has been benchmarked.

**Questions:**

1) For MedExpQA, why not use a translated version of the data?

2) For the Search Query Generation, what is the quality of translations, especially for low-resource languages?

3) What is the impact of the translation?

4) Why did you finally stop at English benchmarks, when there are now other resources in other languages ​​in MCQA? The latter being real data and not translated.

---

> ### Author Response · Authors · 2025-11-25
>
> We would like to thank the reviewer for the feedback.
>
> * We used MedGemma in our experiments; however, we didn’t report results from other medical models since none of the medicine-specific models yielded results comparable to the base models that we used, except MedGemma. The limited language coverage of the open-source models such as Apollo and OLMo was the reason not to include them in our experimental setup.
>
>
> Answers to the questions:
> 1. *For MedExpQA, why not use a translated version of the data?*
> The reasons are: the MedExpaQA repository consists of 32 documents for each question, which sums to 4000 documents in total to translate which is too expensive and complex as it requires medical expertise for each language.
>
> 2. *For the Search Query Generation, what is the quality of translations, especially for low-resource languages?*
> Since the search queries are short texts that require factual information, they usually consist of medical terms which in many cases use shared universal terminology across languages, even in different writing scripts. Therefore, the quality of the queries is high.
>
> 3. *What is the impact of the translation?*
>     * Basque, English, Italian and French translations were manually validated. Kazakh was not. As it is not possible to validate the corpus in such a short period of time, we decided to measure the quality of the automatic translations using backtranslation. Although automatic evaluation of machine translation remains an open research field, we apply backtranslation, one of the methods currently being investigated to automatically measure the quality of machine-translated text. Backtranslation involves translating text from a source language to a target language, then translating it back to the source language and comparing the result with the original to assess translation quality. The comparison is usually done by applying neural metrics, including COMET or BERTScore, although modern lexical metrics such as chrF can also be used.
>
>     * We apply these ideas to our use case and perform backtranslation from the original human-written questions in Spanish to the other languages and back. As it can be seen in the Table below, the results from backtranslation are consistently high across all languages. BERTScore values of 0.90-0.96 indicate that 90-96% of the original meaning is preserved, while COMET scores 0.83-0.86 reflect professional-level surface fidelity that aligns with human “good” translation judgments.
>
>     * Kazakh has lower scores across all metrics, but this is unlikely to be a case of translationese, as the overlap lexical metrics (ChrF and ChrF++) would usually be further inflated. More importantly, high-resource languages like English, French, and Italian exhibit nearly identical translation quality (BERTScore: 0.95-0.96), yet still show notable performance differences, another argument indicating that translation is not the primary driver of the effects we observe.
>    |  | BERTScore | COMET | ChrF | ChrF++ |
>    |---|---|---|---|---|
>    | ES -> EN -> ES | 0.9498 | 0.8538 | 83.48 | 82.05 |
>    | ES -> IT -> ES   | 0.9646 | 0.8643 | 88.45 | 87.58 |
>    | ES -> FR -> ES | 0.9513 | 0.8564 | 84.61 | 83.23 |
>    | ES -> KZ -> ES | 0.9013 | 0.8349 | 71 | 68.75 |
>    | ES -> EU -> ES | 0.9437 | 0.8549 | 82.22 | 80.7 |
>
> 4. *Why did you finally stop at English benchmarks, when there are now other resources in other languages ​​in MCQA? The latter being real data and not translated.*
> Thanks for the suggestion, we have been made aware of the existence of other multilingual data after submission to ICLR.

---

### Official Review · Reviewer_p447 · 2025-11-01

**Soundness:** 2
**Presentation:** 3
**Contribution:** 2
**Rating:** 4
**Confidence:** 3

**Summary:**

This paper investigates multilingual medical question answering by evaluating how different types of external evidence (local repositories, web search, and LLM-generated content) affect performance across six languages with varying resource availability. The authors test multiple LLMs of different sizes and report that larger models generally perform better, English-based retrieval often outperforms target-language retrieval even for non-English queries, and external knowledge integration shows diminishing or negative returns for larger models. While the empirical scope is broad and the multilingual focus is valuable, the paper suffers from methodological concerns, limited explanatory depth, and conclusions that largely confirm expected patterns without providing novel insights into the underlying mechanisms.

**Strengths:**

1. This paper proposes to address an essential real-world problem, medical QA in low-resource languages.
2. The authors conduct a broad and systematic comparison across languages, retrieval methods, and model scales.
3. Some findings (e.g., English retrieval often helps non-English queries; diminishing returns of retrieval for large models) are practically informative.
4. Generally clear structure and presentation, with use of multiple benchmark datasets.

**Weaknesses:**

1. Query generation uses Llama-3.3-70B-Instruct while evaluation uses Llama-3.1-70B; these are nearly identical models, risking bias toward Llama-family strengths.

2. Basque and Kazakh data are machine-translated without expert verification, undermining the validity of “multilingual” claims.

3. Different methods provide unequal amounts of evidence (5 vs 10 vs 32 docs), so results may reflect information quantity, not retrieval quality.

4. The title’s “size not language” framing ignores that language effects remain strong, especially for low-resource languages.

5. No confidence intervals or significance tests despite small test size (125 items).

6. Paper reports patterns (e.g., retrieval hurts large models) but gives no causal insight into why.

**Questions:**

See weakness

---

> ### Author Response · Authors · 2025-11-25
>
> We thank the reviewer for the careful assessment of our work and for the constructive feedback. We address and clarify each point below:
>
> * *Query generation uses Llama-3.3-70B-Instruct while evaluation uses Llama-3.1-70B; these are nearly identical models, risking bias toward Llama-family strengths.*
>   Our results indicate that there is no bias towards Llama models from query generation. Qwen is the best performing model across different scales and if there was any bias from the Llama-family models, then we would have observed a trend where those outperform the others. This is not the case based on our results.
>
> * *Validity of "multilinguality" claims:*
>
> Basque, English, Italian, and French translations were manually validated, whereas the Kazakh translations were not. Due to time constraints, manual validation of the entire Kazakh corpus was not feasible.
> Automatic evaluation of machine translation remains an active area of research and backtranslation represents one of the methods under investigation for quality assessment of machine-translated content. The backtranslation approach involves translating text from a source language into a target language, then retranslating it back into the source language, and quantitatively comparing the backtranslated output with the original text. This comparison is typically performed using neural similarity metrics such as COMET or BERTScore, although modern lexical metrics, including chrF, may also be applied.
>
> We adapted this methodology to our specific use case by backtranslating the original human-authored Spanish questions into each target language and then back into Spanish. As shown in the table below, backtranslation scores remained consistently high across all language pairs. BERTScore values ranging from 0.90 to 0.96 suggest that 90-96% of semantic content is preserved during the translation cycle, while COMET scores of 0.83-0.86 correspond to professional-grade translation quality that typically aligns with human assessments of "good" translations.
>
> Kazakh exhibited lower scores across all metrics; however, this pattern is unlikely to reflect translationese effects, as lexical overlap metrics (ChrF and ChrF++) would typically show inflated values in the presence of translationese. Notably, high-resource languages (English, French, and Italian) demonstrated nearly equivalent translation quality scores (BERTScore: 0.95-0.96), yet still exhibited measurable performance differences in downstream tasks. This finding provides additional evidence that translation quality is not the primary factor driving the observed effects in our study.
> |  | BERTScore | COMET | ChrF | ChrF++ |
> |---|---|---|---|---|
> | ES -> EN -> ES | 0.9498 | 0.8538 | 83.48 | 82.05 |
> | ES -> IT -> ES   | 0.9646 | 0.8643 | 88.45 | 87.58 |
> | ES -> FR -> ES | 0.9513 | 0.8564 | 84.61 | 83.23 |
> | ES -> KZ -> ES | 0.9013 | 0.8349 | 71 | 68.75 |
> | ES -> EU -> ES | 0.9437 | 0.8549 | 82.22 | 80.7 |
>
> * *Information quantity vs. retrieval quality.* Actually, the best performing method from our experiments is the web-search with 10 documents, compared to 32 from the local stores from which we conclude that information quality is more important than the amount of information passed.
>
> * *The title’s “size not language”* entails that in order to get fair results in different languages, it is better to use the models with the highest number of parameters instead of doing retrieval.
>
> * **The statistical significance** of each external knowledge source wrt baseline for English (we obtained the results for all the languages and all of them have p-value < 0.001, for brevity's sake we provide the results from English)
> | EN | MedExpQA | | Websearch | | Generation | |
> |---|---|---|---|---|---|---|
> | | χ^2 | p-value | χ^2 | p-value | χ^2 | p-value |
> | Llama-3.1-8B-Instruct | 55.92 | 7.55e-14 | 27.3 | 1.74e-07 | 42.69 | 6.42e-11 |
> | Qwen3-8B | 46.66 | 8.45e-12 | 38.19 | 6.40e-10 | 41.36 | 1.26e-10 |
> | Gemma-3-12B-it | 27.01 | 2.03e-07 | 14.6 | 1.33e-04 | 33.33 | 7.76e-09 |
> | Qwen3-14B | 62.65 | 2.47e-15 | 58.79 | 1.76e-14 | 65.31 | 6.40e-16 |
> | Gemma-3-27B-it | 22.49 | 2.11e-06 | 10.71 | 1.07e-03 | 88.78 | 4.41e-21 |
> | MedGemma-3-27B-it | 50.89 | 9.76e-13 | 29.15 | 6.70e-08 | 106.98 | 4.51e-25 |
> | Qwen3-32B | 32.76 | 1.04e-08 | 29.52 | 5.55e-08 | 41.79 | 1.01e-10 |
> | Llama-3.1-70B-Instruct | 35.01 | 3.27e-09 | 56.29 | 6.26e-14 | 41.63 | 1.10e-10 |
> | Qwen2.5-72B-Instruct | 58.64 | 1.89e-14 | 50.66 | 1.10e-12 | 59.72 | 1.09e-14 |
>
> * *Paper reports patterns (e.g., retrieval hurts large models) but gives no causal insight into why.:*
>  We assume that the main reason for the performance degradation is knowledge conflicts between the information encoded in the model parameters and the retrieved data. For instance, according to https://arxiv.org/pdf/2509.04304 the models would prefer the outdated information that it was trained on rather than the recent and altered information on the same topic.

---

### Official Review · Reviewer_meEp · 2025-11-01

**Soundness:** 2
**Presentation:** 2
**Contribution:** 2
**Rating:** 4
**Confidence:** 5

**Summary:**

**Problem and Motivation:**
The paper addresses a significant gap in medical LLM research, which is overwhelmingly concentrated on English. This English-centric focus creates two primary problems: (1) It is difficult to generalize findings or apply models to other languages, and (2) nearly all expert medical knowledge bases (e.g., PubMed, StatPearls) are in English, making it unclear how to provide accurate, evidence-grounded answers for medical queries in other languages.
While scaling LLMs has shown they can memorize a substantial amount of domain knowledge (parametric knowledge), it is unknown how this internal knowledge interacts with external evidence, especially in a multilingual context. This work investigates whether LLMs' parametric knowledge is sufficient for multilingual medical QA or if external knowledge is required, and how the language of that external knowledge impacts performance.

**Method and Research Questions:**
This paper presents a systematic investigation of Multilingual Medical Question Answering (MMQA) across both high-resource (English, Spanish, French, Italian) and low-resource (Basque, Kazakh) languages.
The core methodology is to evaluate LLMs of varying sizes (from 8B to 72B parameters) under different knowledge conditions. The authors compare model performance using only internal (parametric) knowledge against performance when augmented with three distinct types of external evidence.
The study is guided by three main research questions:
- RQ1: Is there a single method (i.e., knowledge source) that consistently performs best across all languages?
- RQ2: How do retrieval quality and generation accuracy compare when using English as the evidence source versus using evidence in the target language?
- RQ3: Do LLMs encode sufficient internal (parametric) knowledge for optimal performance, or is external knowledge necessary?

**Knowledge Sources Investigated:**
The authors evaluate three distinct sources of knowledge to augment the LLMs:
- Local Knowledge Repositories: This uses a static, pre-existing English corpus from the MedExpQA benchmark. This corpus includes content from PubMed, Wikipedia, StatPearls, and medical textbooks. The top 32 relevant documents are retrieved for each question.
- Dynamic Web Search: This source uses dynamically retrieved web content. The authors use two different search APIs (Cohere and Google Search via Serper) to find relevant snippets. Retrieval is performed both in English and in the specific target language (e.g., Basque) for comparison.
- Parametric (LLM-Generated) Knowledge: This strategy tests the model's own internal knowledge. An LLM (Llama-3.3-70B-Instruct) is prompted to generate explanations and answers to the query without any external documents. This generated text is then used as the "evidence" for the final answer generation step.

**Experimental Setup:**
- *Dataset:* The study uses the test set (125 questions) from the CasiMedicos dataset, a multilingual, parallel collection of medical exam questions.
- *Languages:* The evaluation covers 6 languages; 4 high-resource (English, Spanish, French, Italian) and 2 low-resource (Basque, Kazakh). The low-resource versions were created for this study via translation with Claude-4-Sonnet.
- *Models:* 9 models from 3 families are tested: Qwen: 8B, 14B, 32B (Qwen 3), 72B (Qwen 2.5); LLaMA: 8B, 70B (Llama 3.1); Gemma: 12B, 27B (Gemma 3), and MedGemma (27B)

**Results & Key Findings:**
- Finding 1 (Best Overall Strategy): The most effective and consistent strategy across all languages (including high and low-resource) was using LLMs augmented with web-retrieved documents in English.
- Finding 2 (English Superiority): Retrieving evidence in English consistently surpassed the performance of language-specific retrieval.
- Finding 3 (Model Scale is Key): Larger models (e.g., Llama 70B, Qwen 72B) consistently outperformed smaller models in all settings.
- Finding 4 (RAG is Not Always Better): The paper challenges the idea that RAG is a uniform improvement. For large models (e.g., >70B) answering in high-resource languages, adding external knowledge (especially from static repositories or LLM-generated explanations) often degraded performance compared to the baseline (no retrieval). This suggests their strong internal (parametric) knowledge conflicts with or is not improved by the provided evidence.
- Finding 5 (Static Repositories are Insufficient): Locally stored, "authoritative" repositories like MedExpQA (PubMed, etc.) provided the least benefit. The results indicate these sources have incomplete domain coverage and inadequate multilingual support compared to dynamic web retrieval.

**Strengths:**

- The paper addresses an important problem: the intersection of multilinguality, model scaling, and external evidence integration for medical QA.
- The experimental design spans several languages and systematically compares static (MedExpQA), dynamic (web search), and parametric (LLM-generated) knowledge sources across a range of open-source LLMs of varying sizes.
- The work is data-driven, employing 6 languages, both high-resource and low-resource, moving beyond the more commonly explored English/French/Spanish scope.

**Weaknesses:**

- *Too small test set.* The paper's conclusions are drawn from a test set of only 125 questions. While the reviewer understands the difficulty of finding multilingual parallel medical QA benchmarks, this sample size is insufficient for making broad, generalizable claims, making the results highly susceptible to variance.
- *Use of unverified machine-translated data.* The low-resource languages (Basque, Kazakh) are not natively sourced but are "silver" translations from Claude-4-Sonnet. This introduces a major confounding variable: it is impossible to distinguish between a model's poor performance due to the language being low-resource and its poor performance due to processing potentially flawed, "unnatural" synthetic translation artifacts. The lack of any human verification or quality analysis of these translations, despite the small test set size, undermines the low-resource language findings.
- *Incomplete related work.* The investigation of "LLM-generated explanations" as a knowledge source is presented without a thorough comparison to existing, formal "generate-then-read" approaches in the medical domain (e.g., MedGENIE@ACL24). This omission weakens the framing of this part of the methodology.
- *Limited and asymmetrical exploratioin of knowledge sources.* The parameters for each knowledge source are narrowly explored. For instance, "Local Knowledge" is limited to the top 32 documents and, most importantly, is sourced only from an English corpus (MedExpQA). This creates an asymmetrical comparison, as it's impossible to know if the failure of "local" knowledge is due to it being static or due to the language mismatch. A fair comparison would have required testing local repositories in their respective target languages.
- *Retriever.* The retrieval similarity metric(s) for static document retrieval (Section 3.1). Is it cosine similarity, dense retrieval, BM25, or hybrid? How are top-k documents scored?
- *Decoding strategies.* The decoding parameters (e.g., temperature, top-p, greedy vs. sampling) for the LLMs are not specified, which is a critical variable in generation tasks.
- *Query generation.* The process for generating queries for the "WebSearch" and "Parametric" sources is not fully detailed.
- *Insufficient detail on statistical significance.* While standard deviations are reported (Table 2), there is limited description of test conditions, repeated runs, or variance across random seeds. The claim that certain results are “statistically significant” (e.g., observed boosts in specific configurations) cannot be robustly validated without description of experimental rigor.
- *Scope and generalizability.* The experiments, while broad linguistically, are focused on multiple-choice QA over clinical cases. It remains unclear how findings generalize to other medical tasks (e.g., summarization, free-form question answering, or patient-facing dialogue)—limiting the broader applicability for some portions of the medical AI community.
- *Absence of error analysis or qualitative case studies.* The paper’s narrative emphasizes quantitative gains or losses (see Table 2), but provides no systematic error dissection. There is a missed opportunity to illustrate the nature of “failure cases”—e.g., what types of questions in Basque or Kazakh are hurt by cross-lingual retrieval, or how content retrieved in a non-target language affects answer quality in practice. Likewise, no real qualitative or visual breakdown is included, undermining claims about “mismatched” or “noisy” evidence.
- *Presentation quality.* The paper's presentation quality is below the expected standard, with low-resolution figures, acronyms introduced multiple times, missing spaces before or after punctuation, wrong citation formats (\citet instad of \citep), etc.

**Questions:**

- Are results across Table 2 and Table 3 statistically significant under repeated runs or across random seeds? Could the authors clarify the number of repeated trials and provide confidence intervals or statistical testing for the main outcomes?
- While the authors state the code and data will be released, they fail to specify a license, which is a requirement for open-sourcing. What is the planned license for the public release of the code and resources?

---

> ### Author Response · Authors · 2025-11-25
>
> We first would like to address and clarify each point listed in the weaknesses:
>
> * *Too small test set*
>   - **125 documents fall well within the range of previously published medical QA research.** Taking this into account, and the fact that we have established statistical significance by means of a chi-square test of independence, our evaluation is methodologically sound and aligns with established precedents in the medical NLP literature. For example, the MMLU datasets, which are often used for comparison, in the Open Medical LLM Leaderboard in HuggingFace (https://huggingface.co/spaces/openlifescienceai/open_medical_llm_leaderboard), contain similar number of instances:
>
>
>     | MMLU Subset | Number of Questions |
>     | --- | --- |
>     | Clinical Knowledge | 265 (Hugging Face) |
>     | Medical Genetics | 100 (Hugging Face) |
>     | Anatomy | 135 (Hugging Face) |
>     | Professional Medicine | 272 (Hugging Face) |
>     | College Biology | 144 (Hugging Face) |
>     | College Medicine | 173 (Hugging Face) |
>
>     Other resources similar to ours in size include, among others:
>
>     https://www.nature.com/articles/s41597-025-05233-z
>
>     https://medvidqa.github.io/
>
>     https://aclanthology.org/2020.nlpcovid19-acl.18/
>
> * *Use of unverified machine-translated data:*
>    * Basque, English, Italian, and French translations were manually validated, whereas the Kazakh translations were not. Due to time constraints, manual validation of the entire Kazakh corpus was not feasible.
> Automatic evaluation of machine translation remains an active area of research and backtranslation represents one of the methods under investigation for quality assessment of machine-translated content. The backtranslation approach involves translating text from a source language into a target language, then retranslating it back into the source language, and quantitatively comparing the backtranslated output with the original text. This comparison is typically performed using neural similarity metrics such as COMET or BERTScore, although modern lexical metrics, including chrF, may also be applied.
> We adapted this methodology to our specific use case by backtranslating the original human-authored Spanish questions into each target language and then back into Spanish. As shown in the table below, backtranslation scores remained consistently high across all language pairs. BERTScore values ranging from 0.90 to 0.96 suggest that 90-96% of semantic content is preserved during the translation cycle, while COMET scores of 0.83-0.86 correspond to professional-grade translation quality that typically aligns with human assessments of "good" translations.
> Kazakh exhibited lower scores across all metrics; however, this pattern is unlikely to reflect translationese effects, as lexical overlap metrics (ChrF and ChrF++) would typically show inflated values in the presence of translationese. Notably, high-resource languages (English, French, and Italian) demonstrated nearly equivalent translation quality scores (BERTScore: 0.95-0.96), yet still exhibited measurable performance differences in downstream tasks. This finding provides additional evidence that translation quality is not the primary factor driving the observed effects in our study.
>     |  | BERTScore | COMET | ChrF | ChrF++ |
>     |---|---|---|---|---|
>     | ES -> EN -> ES | 0.9498 | 0.8538 | 83.48 | 82.05 |
>     | ES -> IT -> ES   | 0.9646 | 0.8643 | 88.45 | 87.58 |
>     | ES -> FR -> ES | 0.9513 | 0.8564 | 84.61 | 83.23 |
>     | ES -> KZ -> ES | 0.9013 | 0.8349 | 71 | 68.75 |
>     | ES -> EU -> ES | 0.9437 | 0.8549 | 82.22 | 80.7 |
>
> * *Incomplete related work.*
> Thanks for pointing out the missing related paper. We will add it.
>
> * *Limited and asymmetric knowledge source.*
>     1. 32 documents were drawn from the previous work in MedRAG/MedExpQA/etc. papers. They concluded that retrieving and using the top 32 closest documents is sufficient for answering medical questions.
>     2. As pointed out in Table 1a, a disproportion in resources is a universal problem as there are no sufficient knowledge sources equivalent to English. Creating equivalent local repositories for each language would require manually curating medical literature that largely does not exist for lower-resource languages. This scarcity is precisely what our paper aims to highlight.
>    3. While we cannot fully disentangle language mismatch from static nature, our results provide suggestive evidence:
>        Table 2 shows that even for Spanish (the original language of the dataset):
>
>          * MedExpQA (English): 75.16% accuracy
>          * Web search (Spanish): 75.64% accuracy
>          * Web search (English): 79.02% accuracy
>
> The similar performance between English MedExpQA and Spanish web search suggests that language mismatch alone cannot explain MedExpQA's limitations. The English web search outperforming both indicates that content quality/coverage matters more than language matching.

---

> > ### Author Response · Authors · 2025-11-25
> >
> > * *Retriever:*
> > The retrieved 32 documents are the result of a combination of using BM25 and MedCPT.
> >
> > * The statistical significance of each external knowledge source wrt baseline for English (we obtained the results for all the languages and all of them have p-value < 0.001, for brevity's sake, we provide the results from English):
> > | EN | MedExpQA | | Websearch | | Generation | |
> > |---|---|---|---|---|---|---|
> > | | χ^2 | p-value | χ^2 | p-value | χ^2 | p-value |
> > | Llama-3.1-8B-Instruct | 55.92 | 7.55e-14 | 27.3 | 1.74e-07 | 42.69 | 6.42e-11 |
> > | Qwen3-8B | 46.66 | 8.45e-12 | 38.19 | 6.40e-10 | 41.36 | 1.26e-10 |
> > | Gemma-3-12B-it | 27.01 | 2.03e-07 | 14.6 | 1.33e-04 | 33.33 | 7.76e-09 |
> > | Qwen3-14B | 62.65 | 2.47e-15 | 58.79 | 1.76e-14 | 65.31 | 6.40e-16 |
> > | Gemma-3-27B-it | 22.49 | 2.11e-06 | 10.71 | 1.07e-03 | 88.78 | 4.41e-21 |
> > | MedGemma-3-27B-it | 50.89 | 9.76e-13 | 29.15 | 6.70e-08 | 106.98 | 4.51e-25 |
> > | Qwen3-32B | 32.76 | 1.04e-08 | 29.52 | 5.55e-08 | 41.79 | 1.01e-10 |
> > | Llama-3.1-70B-Instruct | 35.01 | 3.27e-09 | 56.29 | 6.26e-14 | 41.63 | 1.10e-10 |
> > | Qwen2.5-72B-Instruct | 58.64 | 1.89e-14 | 50.66 | 1.10e-12 | 59.72 | 1.09e-14 |

---

### Official Review · Reviewer_eTye · 2025-11-04

**Soundness:** 2
**Presentation:** 2
**Contribution:** 2
**Rating:** 2
**Confidence:** 4

**Summary:**

The paper investigates the use of RAG for medical question answering across multiple languages. The paper uses an existing benchmark and automatically translates it into two different, low-resource, languages. The paper experiments with different methods to augment the questions with external information, such as CoT, web search, pubmed articles etc. The results do not appear very conclusive and expectedly, model size seems to be the best performance predictor, CoT and RAG across similar-sized models and even languages affect the performance marginally.

**Strengths:**

Studies on low-resource languages are generally important.

**Weaknesses:**

The contribution of the paper does not feel substantial - in the end, there is only one experiment, in which QA performance of 9 LLMs is evaluated with various prompting techniques (bar the results reported in Table 3, but they seem tangential to the paper's core investigation)

The results are inconclusive, and I also don't trust the differences given the small size of the test set - 125 questions. Reported results should be complemented by statistical significance analyses to confirm that the findings are indeed significant. Furthermore, I don't think that automatically translating the resources in other languages shows much more other than the effects of ["translationese"](https://www.cambridge.org/core/journals/natural-language-processing/article/emerging-trends-translationese/D39ADC5B44B06358A153F8926F88DD93), especially since the translations seems to not have been validated by L1/L2 speakers of said languages. The paper claims that "retrieving information in English for non-English questions" is better than retrieving information in the original language. Given that the "original language" questions are translated from English using an LLM, i am not at all surprised by that finding.

The writing is bad. Many claims are asserted without evidence, e.g. in lines 92-93: "common assumption in knowledge-augmented medical Question Answering: that adding external evidence uniformly improves performance". Who assumes that? Similarly, most of pages 7-9 is text, that makes no references to existing literature or other figures/tables, thus making it very hard to distinguish from being generated by an LLM.

**Questions:**

Please provide statistical significance tests to your results and substantiate your claims either by further analyses or by reference to existing literature.

**Details Of Ethics Concerns:**

Not sure if the original dataset was allowed to be run through data-retaining APIs such as anthropic.

---

> ### Author Response · Authors · 2025-11-25
>
> We thank the reviewer for the careful assessment of our work and for the constructive feedback. We address and clarify each point below:
>
> - **“The contribution of the paper does not feel substantial - in the end, there is only one experiment, in which QA performance of 9 LLMs is evaluated with various prompting techniques.”**
> Our contribution is not a single experiment but a systematic multilingual evaluation spanning 6 languages (4 high- and 2 low-resource), 3 types of external evidence sources, and 2 different retrieval languages (EN vs in-language). We would like to emphasize:
>     - No prior work has evaluated and benchmarked multilingual medical QA across diverse languages using multiple external knowledge sources (RAG, local repositories, LLM-generated evidence).
>     - We contribute the first analysis of cross-lingual retrieval in medical QA, showing that retrieval in English outperforms same-language retrieval, even for non-English queries.
>     - Our multilingual extension to Basque and Kazakh, while based on automated translation, constitutes the first low-resource multilingual medical QA evaluation with RAG across real-world knowledge sources.
> - **“Results appear inconclusive; model size is the main predictor.”** The goal of our paper is not to state that “the bigger model the better”, but to empirically establish the optimal strategy to perform Medical QA in a multilingual setting, including parametric and external knowledge. Interestingly, the results of our work show that incorporating external evidence hurts the large models' performance, especially in English. The main takeaways from our study are:
>     - Small models (<30B): Always use English web retrieval
>     - Large models (>70B): Retrieval may decrease performance
>     - Low-resource settings: Either scale model OR use English retrieval
>     - **Web search is preferable to static repositories across** ALL model-language combinations
>     - **English retrieval is preferable to any retrieval in the target language** even for original Spanish (79.02% vs 75.64%)
>     - **Retrieval helps small models, but hurts large ones**:
>         - 8-14B models: +8.2% with retrieval
>         - 70B+ models: -2.4% with retrieval
>
>     These conclusions from the paper mean that the "size is the main predictor" statement does not hold. Furthermore, the results are not inconclusive. We performed a chi-square statistical test of independence between each of the experiments and the baseline (just using the LLM as it is in zero-shot), and all the results obtained are highly significant at the 0.001 level.
>
> * **“Small test set (125 questions); significance tests missing.”**
> The statistical significance of each external knowledge source wrt baseline for English (we obtained the results for all the languages and all of them have p-value < 0.001, for brevity's sake we provide the results from English)
> | EN | MedExpQA | | Websearch | | Generation | |
> |---|---|---|---|---|---|---|
> | | χ^2 | p-value | χ^2 | p-value | χ^2 | p-value |
> | Llama-3.1-8B-Instruct | 55.92 | 7.55e-14 | 27.3 | 1.74e-07 | 42.69 | 6.42e-11 |
> | Qwen3-8B | 46.66 | 8.45e-12 | 38.19 | 6.40e-10 | 41.36 | 1.26e-10 |
> | Gemma-3-12B-it | 27.01 | 2.03e-07 | 14.6 | 1.33e-04 | 33.33 | 7.76e-09 |
> | Qwen3-14B | 62.65 | 2.47e-15 | 58.79 | 1.76e-14 | 65.31 | 6.40e-16 |
> | Gemma-3-27B-it | 22.49 | 2.11e-06 | 10.71 | 1.07e-03 | 88.78 | 4.41e-21 |
> | MedGemma-3-27B-it | 50.89 | 9.76e-13 | 29.15 | 6.70e-08 | 106.98 | 4.51e-25 |
> | Qwen3-32B | 32.76 | 1.04e-08 | 29.52 | 5.55e-08 | 41.79 | 1.01e-10 |
> | Llama-3.1-70B-Instruct | 35.01 | 3.27e-09 | 56.29 | 6.26e-14 | 41.63 | 1.10e-10 |
> | Qwen2.5-72B-Instruct | 58.64 | 1.89e-14 | 50.66 | 1.10e-12 | 59.72 | 1.09e-14 |
>
> - Regarding the size of the test, **125 documents fall well within the range of previously published medical QA research.** Taking this into account, the fact that we have established statistical significance with a chi-square test of independence, our evaluation is methodologically sound and aligns with established precedents in the medical NLP literature, for example, the MMLU datasets, which are often used for comparison, for example, in the Open Medical LLM Leaderboard in HuggingFace (https://huggingface.co/spaces/openlifescienceai/open_medical_llm_leaderboard)
>
>
>     | MMLU Subset | Number of Questions |
>     | --- | --- |
>     | Clinical Knowledge | 265 (Hugging Face) |
>     | Medical Genetics | 100 (Hugging Face) |
>     | Anatomy | 135 (Hugging Face) |
>     | Professional Medicine | 272 (Hugging Face) |
>     | College Biology | 144 (Hugging Face) |
>     | College Medicine | 173 (Hugging Face) |
>
>     Other resources similar to ours in size include, among others:
>
>     https://www.nature.com/articles/s41597-025-05233-z
>
>     https://medvidqa.github.io/
>
>     https://aclanthology.org/2020.nlpcovid19-acl.18/

---

> ### Author Response · Authors · 2025-11-25
>
> - **“Translations into low-resource languages are not validated; results reflect translationese.”**.
>
> Basque, English, Italian, and French translations were manually validated, whereas the Kazakh translations were not. Due to time constraints, manual validation of the entire Kazakh corpus was not feasible. Consequently, we employed backtranslation as a proxy measure for assessing the quality of machine-translated text.
>
> Automatic evaluation of machine translation remains an active area of research and backtranslation represents one of the methods under investigation for quality assessment of machine-translated content. The backtranslation approach involves translating text from a source language into a target language, then retranslating it back into the source language, and quantitatively comparing the backtranslated output with the original text. This comparison is typically performed using neural similarity metrics such as COMET or BERTScore, although modern lexical metrics, including chrF, may also be applied.
>
> We adapted this methodology to our specific use case by backtranslating the original human-authored Spanish questions into each target language and then back into Spanish. As shown in the table below, backtranslation scores remained consistently high across all language pairs. BERTScore values ranging from 0.90 to 0.96 suggest that 90-96% of semantic content is preserved during the translation cycle, while COMET scores of 0.83-0.86 correspond to professional-grade translation quality that typically aligns with human assessments of "good" translations.
>
> Kazakh exhibited lower scores across all metrics; however, this pattern is unlikely to reflect translationese effects, as lexical overlap metrics (ChrF and ChrF++) would typically show inflated values in the presence of translationese. Notably, high-resource languages (English, French, and Italian) demonstrated nearly equivalent translation quality scores (BERTScore: 0.95-0.96), yet still exhibited measurable performance differences in downstream tasks. This finding provides additional evidence that translation quality is not the primary factor driving the observed effects in our study.
>
> | | BERTScore | COMET | ChrF | ChrF++ |
> |---|---|---|---|---|
> | ES -> EN -> ES | 0.9498 | 0.8538 | 83.48 | 82.05 |
> | ES -> IT -> ES | 0.9646 | 0.8643 | 88.45 | 87.58 |
> | ES -> FR -> ES | 0.9513 | 0.8564 | 84.61 | 83.23 |
> | ES -> KZ -> ES | 0.9013 | 0.8349 | 71 | 68.75 |
> | ES -> EU -> ES | 0.9437 | 0.8549 | 82.22 | 80.7 |
>
> - Regarding the writing style and lack of figures, we will improve the writing, update the figure quality, and add more citations.
> - Here we suggest other citations relevant to the article’s content:
>     - **Benchmarking Retrieval-Augmented Generation for Medicine**, Xiong et. al 2024: reports up to 18% improvement using RAG in medical QA tasks.
>     - **MKRAG: Medical Knowledge Retrieval Augmented Generation for Medical Question Answering**, Shi et. al 2024: they apply external facts to the questions from MedQA and when they add external medical facts, accuracy goes up (from 44.46% to 48.54%) compared to the model without the external facts.
>     - **Rationale-Guided Retrieval Augmented Generation for Medical Question Answering**, Sohn et. al, 2025: report 6.1% increase when I applied RAG.
> - Pages 7-9 do contain internal references to other sections or tables/figures, including discussion of specific results.

---

### Note · Authors · 2025-12-04

I have read and agree with the venue's withdrawal policy on behalf of myself and my co-authors.